# *Leishmania infantum* xenodiagnosis from vertically infected dogs reveals significant skin tropism

**Breanna M. Scorza**[1], **Kurayi G. Mahachi**[1], **Arin C. Cox**[1¤a], **Angela J. Toepp**[1¤b], **Adam Leal-Lima**[1¤c], **Anurag Kumar Kushwaha**[2], **Patrick Kelly**[3], **Claudio Meneses**[4], **Geneva Wilson**[1¤d], **Katherine N. Gibson-Corley**[5¤e], **Lyric Bartholomay**[6], **Shaden Kamhawi**[4], **Christine A. Petersen**[1] *

**1** Department of Epidemiology, University of Iowa, Iowa City, Iowa, United States of America, **2** Department of Medicine, Banaras Hindu University, Varanasi, UP, India, **3** Department of Microbiology, University of Iowa, Iowa City, Iowa, United States of America, **4** National Institute of Allergy and Infectious Disease, NIH, Rockville, Maryland, United States of America, **5** Department of Pathology, University of Iowa, Iowa City, Iowa, United States of America, **6** Department of Epidemiology, University of Wisconsin, Madison, Wisconsin, United States of America

¤a  Current address: Department of Veterinary Biosciences, Ohio State University, Columbus, Ohio, United States of America
¤b  Current address: EVMS-Sentara Healthcare Analytics and Delivery Science Institute, Eastern Virginia Medical School, Norfolk, Virginia, United States of America
¤c  Current address: Universidade Estadual do Ceará, Fortaleza, CE, Brazil
¤d  Current address: Center of Innovation for Complex Chronic Healthcare, Edward Hines Jr. Veteran Affairs Hospital, Hines, Illinois, United States of America
¤e  Current address: Department of Pathology, Microbiology and Immunology, Vanderbilt University Medical Center, Nashville, Tennessee, United States of America
* christine-petersen@uiowa.edu

**Data Availability Statement:** Additional clinicopathological cohort data including complete blood count and serum chemistry panel results for individual xenodiagnosis subjects is available on

## Abstract

### Background

Dogs are the primary reservoir for human visceral leishmaniasis due to *Leishmania infantum*. Phlebotomine sand flies maintain zoonotic transmission of parasites between dogs and humans. A subset of dogs is infected transplacentally during gestation, but at what stage of the clinical spectrum vertically infected dogs contribute to the infected sand fly pool is unknown.

### Methodology/Principal findings

We examined infectiousness of dogs vertically infected with *L. infantum* from multiple clinical states to the vector *Lutzomyia longipalpis* using xenodiagnosis and found that vertically infected dogs were infectious to sand flies at differing rates. Dogs with mild to moderate disease showed significantly higher transmission to the vector than dogs with subclinical or severe disease. We documented a substantial parasite burden in the skin of vertically infected dogs by RT-qPCR, despite these dogs not having received intradermal parasites via sand flies. There was a highly significant correlation between skin parasite burden at the feeding site and sand fly parasite uptake. This suggests dogs with high skin parasite burden contribute the most to the infected sand fly pool. Although skin parasite load and parasitemia

researchgate.net under Christine Anne Petersen's profile at: https://www.researchgate.net/publication/354935313_Clinicopathological_data_to_match_canine_xenodiagnosis.

**Funding:** This work was funded by R01 TW010500 from the Fogarty International Center, National Institutes of Health. This work was completed while B.M.S. was supported by the University of Iowa Interdisciplinary Immunology Postdoctoral Training Grant T32AI007260 National Institute of Allergy and Infectious Diseases, National Institutes of Health. The funders had no role in study design, data collection and analysis, decision to publish, or preparation of the manuscript.

**Competing interests:** The authors have declared that no competing interests exist.

correlated with one another, the average parasite number detected in skin was significantly higher compared to blood in matched subjects. Thus, dermal resident parasites were infectious to sand flies from dogs without detectable parasitemia.

## Conclusions/Significance

Together, our data implicate skin parasite burden and earlier clinical status as stronger indicators of outward transmission potential than blood parasite burden. Our studies of a population of dogs without vector transmission highlights the need to consider canine vertical transmission in surveillance and prevention strategies.

## Author summary

Sand flies transmit *Leishmania* parasites between infected dogs and humans leading to the life-threatening tropical disease Visceral Leishmaniasis (VL). Identifying which dogs transmit parasites well to sand flies is important to curb disease spread. The offspring of both dogs and humans can also be infected vertically while *in utero*. Despite this, the infectiousness of dogs that receive parasites *in utero* to sand flies has not been thoroughly investigated. Thus, we allowed sand flies to feed on a group of vertically infected dogs at varying stages of VL disease severity and measured sand fly parasite uptake. We found vertically infected dogs were readily able to transmit parasites to the sand flies. Dogs that were most infectious had mild to moderate clinical disease and relatively high levels of parasite infection in their blood and skin. However, the level of skin infection was significantly higher than that observed in the blood, and the skin parasite load had the strongest correlation with sand fly parasite uptake. This implicates the skin may be an underappreciated driver of canine infectiousness to the sand fly vector. In addition, this work highlights that vertically infected dogs are very important parts of the transmission cycle and must be considered in all public health efforts addressing VL.

## Introduction

Visceral leishmaniasis (VL) is caused by protozoan *Leishmania infantum* and *L. donovani* parasites, killing an estimated 30,000 people annually [1]. In *L. infantum* endemic areas, dogs are the primary domestic reservoir driving zoonotic VL [2–4]. Vector transmission by phlebotomine sand flies is the predominant mode of parasite transmission; however, transplacental transmission has been demonstrated in both humans and dogs infected with *L. infantum* [4–8]. In the United States, *L. infantum* is enzootic in hunting dogs and vertical transmission alone maintains infection within this population [9,10]. As human VL is inextricably linked to canine infection, understanding factors underlying transmission from vertically *L. infantum*-infected dogs to sand flies is of utmost public health concern.

 Epidemiological modeling and experimental findings support that a fraction of *L. infantum* infected dogs, termed 'super-spreaders', disproportionately contribute to the pool of infectious sand flies [11,12]. Several groups have applied xenodiagnosis, the gold standard for measuring pathogen transmissibility from an infected host to an insect vector, in an effort to identify risk factors associated with increased canine transmission to sand flies [13–15]. Parasitemia and clinical status are the most commonly identified correlates of infectiousness to sand flies [16–

18]. However, widely variable amounts of parasite uptake after xenodiagnosis between sand flies fed on the same host indicates other microenvironmental factors influence this interaction [19].

Importantly, sand flies are telmophages, feeding from blood pools formed by lacerating mammalian skin with a serrated proboscis [19]. This type of feeding allows parasite uptake from dermal blood vessels, as well as disrupted dermal-resident parasitized cells. Recent findings using xenodiagnosis have implicated canine skin parasite load as a critical correlate of canine infectiousness to sand flies [11,18,20,21].

When sand flies egest *L. infantum* into the skin of naïve canine hosts, parasite depots persist at the skin inoculation site [22]. This finding could indicate skin resident parasites are remnants of previous vector bite-site infections. In contrast, congenital transmission of *L. infantum* occurs via a hematological route. This raises the question of whether vertically infected dogs attain a skin parasite burden high enough to influence transmission in the absence of receiving intradermal parasite inoculations from sand flies. Dermotropism of *L. infantum* in dogs known to be infected vertically has not been investigated. Specifically, it is not known whether the same mechanisms that govern parasite transmission during vector transmission will also dictate parasite localization in vertically infected dogs and their ability to transmit out to the sand fly vector.

To address these questions, we assayed skin parasite burden and performed xenodiagnosis on a cohort of dogs vertically infected with *L. infantum* at various stages of clinical disease. We found significant dermotropism of *L. infantum*, highly correlated with transmission to sand flies. This suggests skin parasite infection is linked to parasite transmission from both congenitally and vector infected dogs.

## Materials and methods

### Ethics statement

All animal use involved in this work was approved by the University of Iowa Institutional Animal Care and Use Committee and was performed under the supervision of licensed and, where appropriate, board-certified veterinarians according to International AAALAC accreditation standards. Canine subjects in this study were donated to the University of Iowa after signed informed consent was obtained.

### Animal cohort

Canine subjects were obtained from a cohort of U.S. dogs where vertically transmitted *L. infantum* is enzootic [2,10,23]. *L. infantum* infection of all dogs occurred naturally *in utero*. The LeishVet clinical scores and demographics of the 16 dogs used in this study are provided (S1 Table).

### Leishmaniosis clinical classification

On the day of xenodiagnosis, a veterinarian conducted a physical exam on each dog for clinical signs of leishmaniosis including low body condition score, dermatitis, coat condition, lethargy, lymphdenomegaly, conjunctivitis, alopecia, cutaneous lesions, pale mucous membranes, or epistaxis. Whole blood and serum were collected. Whole blood was subjected to complete blood count (IDEXX Laboratories Inc.) and DNA isolation using the QIAamp Blood DNA Mini Kit according to manufacturer instructions (Qiagen). Real Time-quantitative PCR (RT-qPCR) for *Leishmania* ribosomal DNA was performed as previously described [24]. Serum was used to perform Dual-Path Platform ® Canine Visceral Leishmaniosis (DPP) serological

analysis (ChemBios) detecting antibodies against recombinant *L. infantum* rK28 antigen. ChemBios Micro Reader was used to determine serological status with a reader value >10 considered as seropositive [25]. Serum chemistry panels were performed (IDEXX Laboratories Inc.), S2 Table. In some cases, urine was analyzed by refractometer. The results of physical exam and these tests were combined to clinically stage leishmaniosis as: subclinical (stage 1), mild (stage 2), moderate (stage 3), or severe disease (stage 4) according to the LeishVet Guidelines [26].

## Sand flies and xenodiagnosis

*Lutzomyia longipalpis* sand flies were reared at the National Institute of Allergy and Infectious Diseases, Laboratory of Malaria and Vector Research insectary as previously described [27]. Adult *Lu. longipalpis* were incubated in a humidified chamber at 26˚C with access to 30% sucrose solution. Sand flies were starved 12 hrs prior to xenodiagnosis. For feeding, ~30–40 female sand flies were gently aspirated along with males at an approximate ratio of 3:1 and transferred into 2-inch diameter custom made polycarbonate feeding cups covered on one side with a fine screen-mesh [18,19,28].

Dogs were sedated with dexmedetomidine (Zoetis Inc.). Heartrate and respiratory rate were monitored throughout procedure. Dogs were placed within an enclosed mesh chamber, and two feeding cups per dog were placed on the axillary region and inner pinna for 30 minutes [18,19,28].

Following feeding, sand flies were incubated for 48 hours at 26˚C in a humidified chamber with access to sucrose solution. Due to institutional requirements, 48hrs was the maximum allowable incubation time post-feeding on infected dogs to minimize risk of infection to staff. After incubation, individual engorged blood-fed female sand flies were separated into Eppendorf tubes containing 100 uL of DNA lysis buffer (Gentra Puregene Tissue Kit, Qiagen) and stored at -20˚C.

## Tissue isolation

Following xenodiagnosis and humane euthanasia per AVMA guidelines, a full necropsy was performed by a board-certified veterinary pathologist. Canine spleen samples were obtained, flash frozen on dry ice, and stored at -80˚C until DNA isolation. Skin biopsies were taken from axillary and pinnal sand fly feeding sites as well as contralateral, unfed sites from each study animal using 6 mm punch biopsies (Integra) and placed into zinc formalin for histology or frozen in saline at -80˚C for nucleic acid isolation.

## DNA isolation and real time-qPCR

For preparations of standard curves, low passage *Leishmania infantum* promastigotes (US/2016/MON1/FOXYMO4), originally isolated from a hunting dog with VL, were cultured in complete hemoflagellate-modified minimal essential medium with 10% heat-inactivated fetal calf serum at 26˚C and stationary phase parasites were enumerated by hemacytometer [29]. For sand fly parasite quantification, control female *Lu. longipalpis* sand flies fed on uninfected rabbit blood, incubated 48hrs, then were spiked with known quantities of *L. infantum* promastigotes before DNA isolation. For *Leishmania* quantification in host tissues, a standard curve was created by spiking known quantities of *L. infantum* promastigotes into uninfected canine blood and normalized by input DNA concentration.

Sand fly and canine sample DNA were isolated with the Gentra Puregene Tissue Kit (Qiagen). For *L. infantum* detection by RT-qPCR, *Leishmania* small-subunit rRNA-specific probe (5'-[6-FAM]-CGGTTCGGTGTGTGGCGCC-MGBNFQ-3') and primers (forward, 5'-

AAGTGCTTTCCCATCGCAACT-3'; reverse, 5'-GACGCACTAAACCCCTCCAA-3') were used as previously described [24]. An exponential regression was constructed from the standard curve and used to convert Ct into parasite equivalents per sand fly or per 2.5 ug mammalian DNA.

### Statistical analyses

Normality of data was assessed using the D'Agostino-Pearson test. For comparisons between tissues obtained from the same subject, Wilcoxon matched-pairs signed rank test was performed. For comparisons between subjects, Kruskal-Wallis ANOVA was performed. When appropriate, Dunn's post-test was used for multiple comparisons. Spearman correlation was computed for correlation analyses. For all analyses, significance is: $^*p \leq 0.05$; $^{**}p < 0.01$; $^{***}p < 0.001$; $^{****}p < 0.0001$.

## Results

### Vertically infected dogs harbor significant skin parasite burden

In *L. infantum* endemic areas with vector predominant transmission, sand flies inoculate parasites into host skin, which then disseminate systemically but leave dermal foci of infection at the original bite site(s) [22]. We sought to investigate whether *L. infantum* parasites obtained *in utero* via the bloodstream which lack these original foci of parasites in the skin would also be dermotropic or remain concentrated in viscera. We analyzed parasite burdens in the blood, spleen, and skin in vertically infected dogs from this cohort at various clinical stages of VL (Fig 1). We utilized the LeishVet guidelines, which consider clinical physical signs and clinico-pathologic laboratory findings, to stage canine VL clinical severity: subclinical (stage 1), mild (stage 2), moderate (stage 3), or severe disease (stage 4) [26].

Real Time-qPCR for *Leishmania* DNA in peripheral blood samples revealed none of the LeishVet stage 1 dogs had detectable parasitemia, while 71.4% of LeishVet stage 2 dogs, and 100% of LeishVet stage 3 and 4 dogs were parasitemic, with an average blood burden of 312, 480, and 380 parasite equivalents/2.5 ug DNA, respectively (Fig 1A).

The spleen is a major target organ of *Leishmania* infection. When analyzing splenic parasite load we observed 50% of LeishVet stage 1, 75% of LeishVet stage 2, and 100% of LeishVet stage 3 and stage 4 dogs had detectable parasitism with average burdens of $1.64 \times 10^3$, $9.98 \times 10^4$,

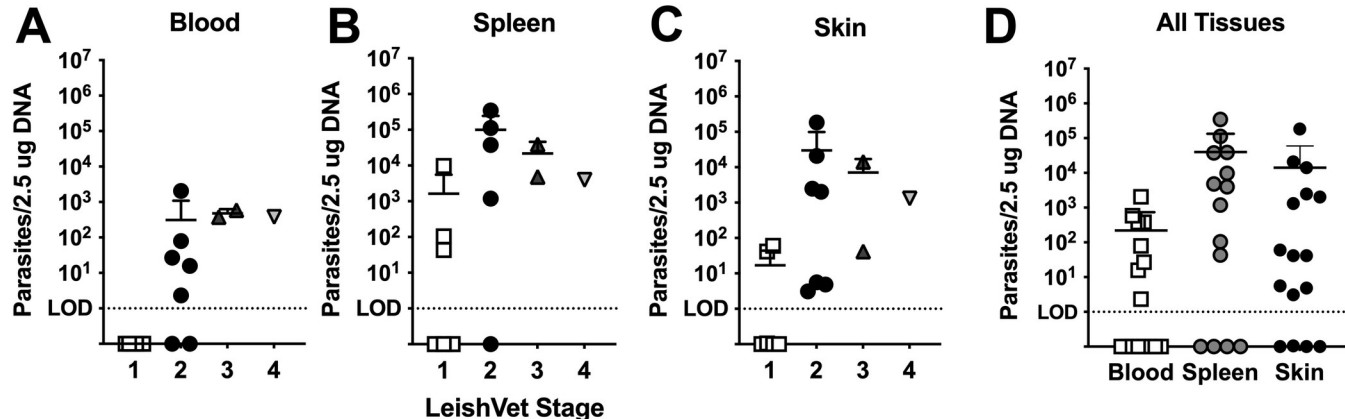

**Fig 1. Vertically infected dogs harbor significant skin *Leishmania* burden.** Calculated *L. infantum* parasite burden in (A) blood (n = 16), (B) spleen (n = 14), (C) skin (n = 16) from dogs at indicated LeishVet clinical stage of disease. (D) Calculated *L. infantum* parasite burden in all tissues from all dogs independent of LeishVet status. Each symbol represents one dog. For skin, the average burden of all sampled skin sites per dog is shown.

$2.2 \times 10^4$, and $4 \times 10^3$ parasite equivalents/2.5 ug DNA, respectively (Fig 1B). As expected, the quantified parasite load in the spleen was higher than observed in the peripheral blood.

To investigate skin parasite burden, we assayed between two and four unique skin biopsy sites per dog. The parasite quantification shown in Fig 1C is the average burden across all biopsy sites/dog (average of 2–4 skin biopsies/dog). Individual skin biopsy burden and parasite burden difference by biopsy location are shown in S2 Fig. In vertically infected dogs, we observed 50% of LeishVet stage 1 and 100% of LeishVet stage 2, stage 3, and stage 4 dogs had detectable skin infection, with average skin burdens of 17.1, $3 \times 10^4$, $7.1 \times 10^3$, and $1.3 \times 10^3$ parasite equivalents/2.5 ug DNA, respectively (Fig 1C). Despite never receiving intradermal parasites from a vector, the skin of vertically infected dogs contained a substantial parasite burden. Parasite burden in each tissue, independent of clinical stage, is shown in Fig 1D.

## Skin parasite burden correlates with systemic parasite load

Parasitemia has long been considered a focal determinant of *Leishmania* vector transmission potential, therefore we were interested in establishing whether skin parasite burden from vertically infected dogs correlated with parasitemia. We found a strong significant positive correlation ($p = 0.002$, $r = 0.75$) between skin parasite burden and corresponding parasitemia (Fig 2A). Despite this, the burden of parasites in the skin was consistently higher than the blood burden detected for every case except one ($p = 0.0046$, Fig 2B). Skin parasite burden also significantly positively correlated with splenic parasite burden ($p = 0.001$, $r = 0.79$, Fig 2C). Moreover, in a paired analysis, skin and splenic parasite load were not significantly different ($p = 0.18$), demonstrating that the skin accumulates parasites to a similar extent as the spleen. This is remarkable, because the spleen is one of the most highly parasitized tissues in VL.

## Dogs with mild to moderate clinical disease show the highest parasite transmissibility to sand flies

Although it was demonstrated that vertically infected dogs could transmit parasites to sand flies [30], it is poorly understood how host factors predict enhanced transmission from this group. For the first time, xenodiagnosis was performed on 16 vertically infected dogs across progressive stages of VL and infection severity. There was no significant difference in the ability of sand flies to feed on dogs (% female sand flies containing blood meal) based on clinical stage (Fig 3A). After feeding, DNA was isolated from individual blood fed sand flies and analyzed against a standard curve to quantify *Leishmania* (S3A Fig).

Although flies successfully fed on all dogs at a similar rate, we measured variable parasite equivalents in each sand fly after feeding. (S3B Fig). Average parasite quantification post-feeding was similar between sand flies fed on pinnae or axillary feeding sites (S3C Fig). Interestingly, the highest percentage of infected sand flies (% of blood fed flies containing >1 parasite equivalent) was found after feeding on LeishVet stage 2 dogs, compared to more clinically apparent stage 3 and 4 dogs (Figs 3B and S3B). Similarly, quantification of parasites taken up by individual sand flies showed dogs at LeishVet stage 2 and 3 took up significantly more parasites than LeishVet stage1, subclinical infection, or stage 4, severe disease, dogs (Fig 3C). As some previous studies have observed increased transmission from dogs classified as symptomatic based on clinical signs of leishmaniosis, we also stratified sand fly parasite uptake by number of leishmaniosis clinical signs observed by physical examination. We found dogs with any clinical signs of disease led to more sand fly parasite uptake than was seen by dogs with zero clinical signs of disease. However, similar to the LeishVet stage, increased parasite uptake did not directly correlate with increasing number of clinical signs (Fig 3D, $p = 0.24$).

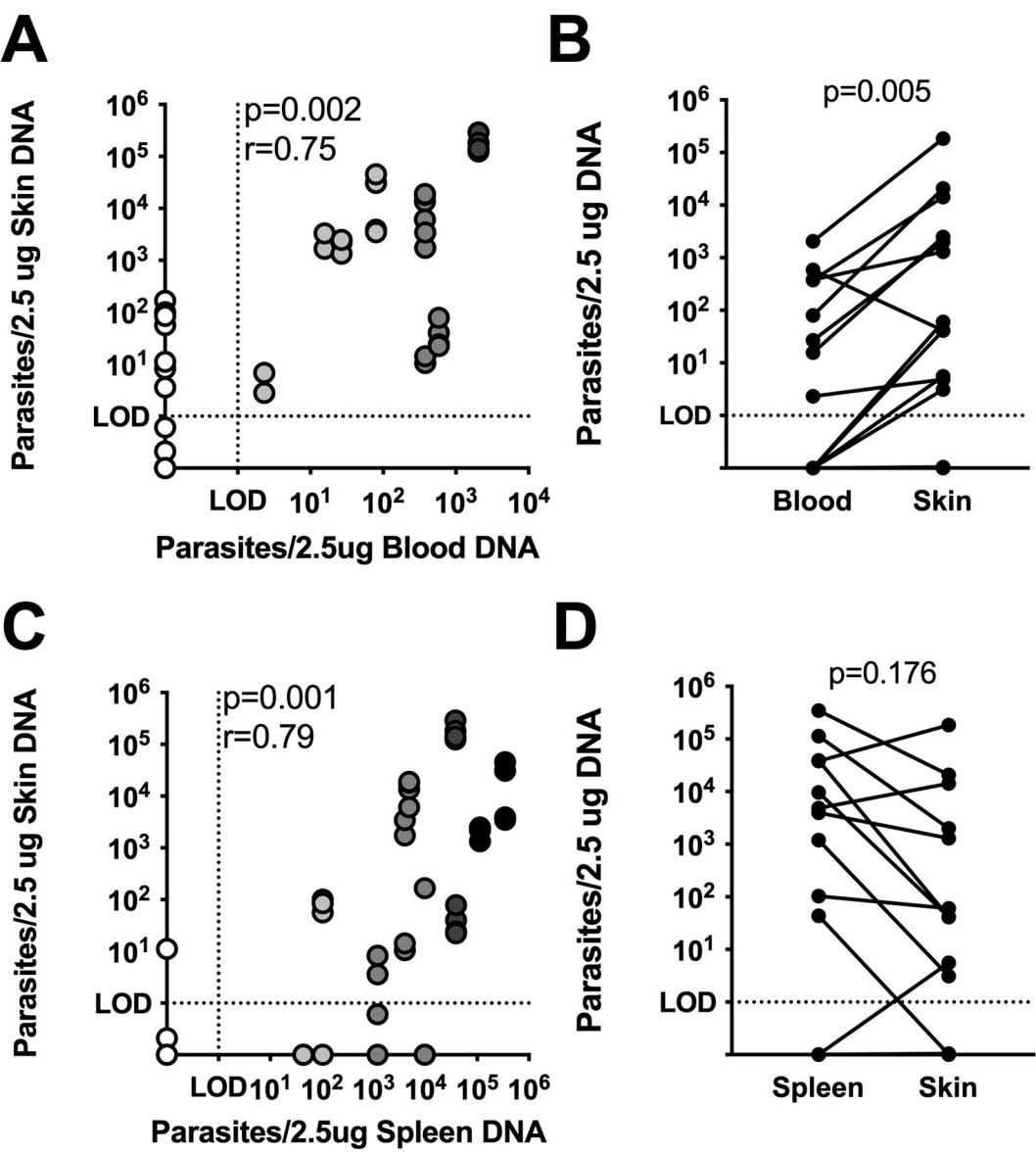

**Fig 2. Skin parasite burden correlates with systemic parasite load.** Spearman's correlation (A) and paired Wilcoxon test (B) between skin parasite load and blood parasite load from the same animals (n = 16). Spearman's correlation (C) and paired Wilcoxon test (D) between skin parasite load and splenic parasite load from the same animal (n = 14). r, Spearman correlation coefficient.

Clinicopathological abnormalities such as signs of non-regenerative anemia, hypergamma-globulinemia, hypoalbuminemia, and increased creatinine are used to inform LeishVet stage categorization. As the sample size in LeishVet stage 3 and 4 was relatively low, we also stratified tissue parasite burden and xenodiagnosis results by whether a dog has progressed to present clinicopathological alterations (S4 Fig). Dogs with clinically apparent leishmaniosis had significantly higher parasite loads measured in their peripheral blood, spleen, and skin tissue compared to those without any clinicopathological abnormalities (S4A–S4C Fig). Although sand flies were able to feed with similar success on dogs with or without abnormal clinicopathological findings, dogs with clinical alterations led to a significantly increased rate of sand fly infection and parasite uptake (S4D and S4E Fig).

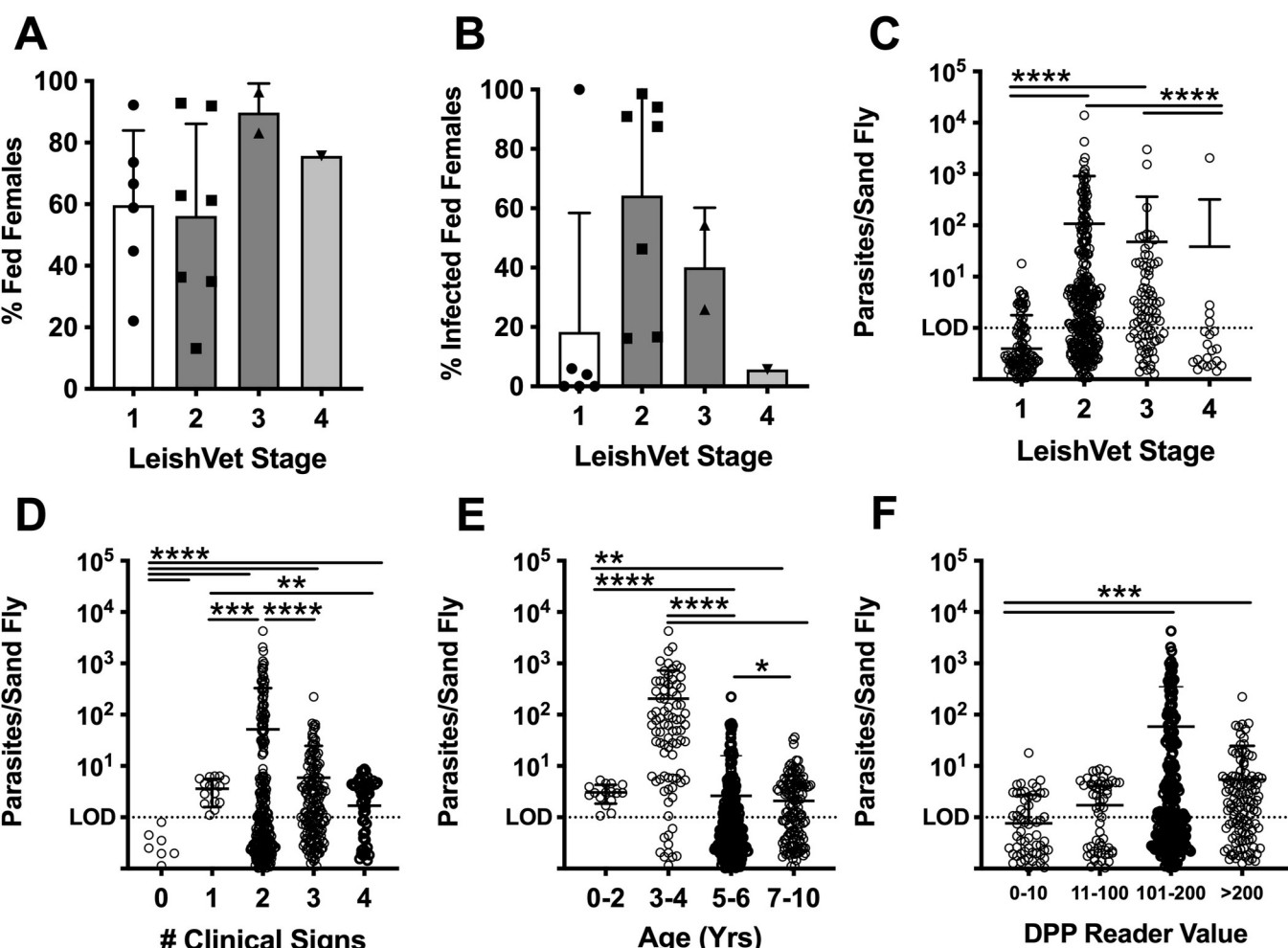

**Fig 3. Higher parasite uptake among sand flies fed on dogs with moderate disease.** (A) Frequency of sand flies containing a blood meal and (B) frequency of blood meal containing sand flies containing >1 calculated parasite after xenodiagnosis on dogs at indicated LeishVet clinical stages. (C-F) Calculated number of parasites per individual sand fly after xenodiagnosis on dogs with indicated LeishVet clinical stage (C), number of VL clinical signs (D), age range (E), or DPP serological value (F). Kruskal Wallis ANOVA with Dunn's post-test. **p<0.01; ***p<0.001; ****p<0.0001.

The age range of dogs with the highest average parasite uptake per sand fly in this cohort was 3–4 years old (Fig 3E). Serological responses to *L. infantum* rK28 antigen rise with increasing disease severity [25], therefore we compared DPP®CVL reader value correlation with parasite uptake by sand flies and found seronegative dogs (Reader Value 0–10) had the lowest transmission to sand flies, while dogs in the two highest serologically positive bins resulted in significantly higher transmission to sand flies (Fig 3F). Therefore, none of these factors (clinical severity, age, or serology) directly correlated with parasite transmission to sand flies. Instead, we observed a "Goldilocks" effect, where dogs with moderate disease resulted in the highest rate of sand fly infection and highest parasite burden in infected sand flies.

## Skin parasite burden is the best correlate of successful parasite transmission to sand flies

Parasitemia is known to correlate with *Leishmania* parasite transmission to sand flies [16–18]. However, in almost every subject, the skin of vertically infected hounds contained a higher

parasite burden compared to matched blood samples (Fig 2B). Therefore, we were very interested to know if skin parasite load in this cohort would correlate with transmission to sand flies. In Fig 4A, there was a strong positive correlation (Spearman r = 0.68, p = 0.005) between skin parasite burden and sand fly parasite uptake. Parasite load in the blood (Spearman r = 0.64, p = 0.009) and spleen (Spearman r = 0.52, p = 0.058) also positively correlated with sand fly parasite uptake. However, the parasite burden in the skin of our cohort showed both the strongest correlation coefficient and the most significant association with parasite transmission to sand flies by xenodiagnosis.

## Discussion

This study is the first investigation into parasite dermotropism and its relationship to xenodiagnosis in a vertically infected canine VL cohort. Our data supports significant parasite burdens in sand fly accessible tissues, such as blood and particularly skin, as the driving factor determining infectiousness of *L. infantum*-infected dogs. We found *L. infantum* accumulated in vertically infected dog skin that were significantly higher than those measured in the blood and approaching parasite burdens found in splenic tissue.

Dermal parasite load also differentiates human hosts contributing to the infectious sand fly pool. In people with post-Kala Azar dermal leishmaniasis (PKDL) due to *L. donovani*, which causes VL in India and east Africa, skin parasite load significantly predicted infectiousness to *Phlebotomus argentipes* sand flies by xenodiagnosis [28,31]. PKDL patients had relatively high skin parasite loads, but blood parasite burdens below the threshold of detection and were still infectious to sand flies via xenodiagnosis [28]. In this study, seropositive but subclinical individuals were not infectious to the vector, similar to our findings of subclinical LeishVet stage 1 dogs. This indicates that it is skin parasite burden, more so than parasitemia, that drives transmissibility of *Leishmania donovani* complex parasites from PKDL patients.

The mechanisms *Leishmania* use to traffic to and seed dermal tissues have not been fully elucidated. It was demonstrated that intravenously administered *Leishmania* parasites trafficked to the skin of B6.*Rag2*[-/-] mice, establishing macro- and microscopic pockets of skin resident infection [19]. This 'patchy' parasite distribution better predicted the variable parasite uptake observed in sand flies after xenodiagnosis compared to mathematical models assuming a homogenous parasite source from blood [19]. Our study biopsied multiple skin sites and found limited variability in skin parasite burden from the same subject at the macro level (S2 Fig). Thus, the microscopic parasite burden at the sand fly bite site seems to influence transmission.

The proficient *Leishmania* dermotropism we found in this population of dogs with vertically transmissible parasites may be evolutionarily advantageous to facilitate vector transmission. Changes in the volatile organic compound profile, or odorous compounds potentially detectable by sand flies, in the hair of dogs from an *L. infantum* endemic area had high sensitivity and specificity for identifying infected dogs [32,33]. Indeed, an olfactometer bioassay found female *Lu. longipalpis* sand flies were significantly more attracted to hamsters with established *L. infantum* infections [34]. Therefore, a high level of parasites in skin may lead to biochemical changes attractive to the vector, enhancing feeding efficiency and transmission.

We saw that by grouping dogs presenting complete blood count or serum chemistry alterations had significantly higher tissue parasite burdens, which lead to increased parasite uptake by xenodiagnosis, compared to dogs with no clinicopathological abnormalities. However, using the higher resolution LeishVet scale to stage disease revealed an interesting dynamic. Mild to moderately diseased dogs (LeishVet levels 2–3) were the most infectious to *Lu. longipalpis* sand flies compared to dogs with subclinical (LeishVet level 1) or severe disease

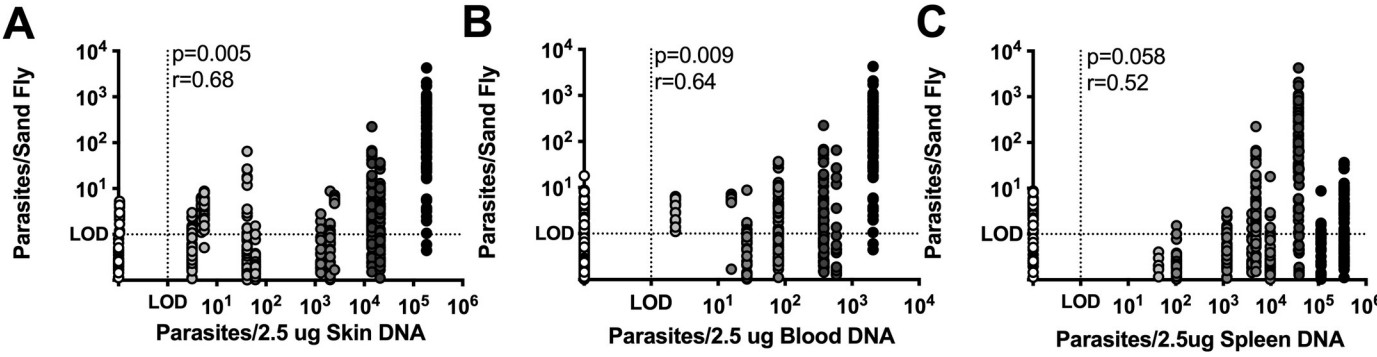

**Fig 4. Skin parasite burden is best correlate of parasite transmission to sand flies.** Correlation between calculated parasite uptake by individual sand flies after xenodiagnosis and paired (A) average skin parasite burden (n = 16), (B) blood parasite burden (n = 16), (C) or splenic parasite burden (n = 14). Skin parasite burden was averaged between the two biopsies collected from ipsilateral feeding sites. Spearman correlation p-value and correlation coefficient (r) are shown.

(LeishVet level 4). This may explain conflicting reports of xenodiagnosis association with clinical signs [13,14,16–18,35,36]. Dogs with moderate disease may fall into different asymptomatic or symptomatic categories depending on the classification system, and this highlights the need for more standardized staging criteria.

It is interesting to consider why parasites from dogs with severe VL disease may not be as transmissible. Here, *Lu. longipalpis* sand flies successfully fed on a LeishVet stage 4 dog but took up relatively few parasites. LeishVet level 3 and 4 dogs were defined by compromised renal function, which can lead to accumulation of nitrogenous blood products such as blood urea nitrogen as well as elevated breakdown products like creatinine (S2 Table) [26]. These waste products may interfere with blood cell integrity, and parasite viability within the sand fly midgut. The sand flies fed on a LeishVet stage 4 dog in this study were observed containing relatively dark, precipitated blood meals after 48 hrs of incubation, which may indicate altered blood meal digestion. Xenodiagnosis of additional dogs in this clinical range would be necessary to confirm this relationship.

Diagnosing VL early in the infectious course in dogs is difficult due to the non-specific nature of initial clinical signs and clinicopathological results. Interestingly we observed several dogs with undetectable parasitemia already had detectable parasites in the skin by qPCR. This could implicate skin biopsy as a more sensitive diagnostic in these instances. Importantly, some sand flies fed on these dogs were able to successfully take up parasites, demonstrating that these early-stage dogs are also infectious to sand flies as seen in our experiments, despite having a low blood burden.

Although we quantified parasite uptake after xenodiagnosis with a high degree of sensitivity utilizing qPCR, a limitation of this study is that parasite viability in the sand fly midgut was not verified by microscopy, and therefore we cannot assume all parasites survived ingestion. Due to institutional requirements, this study was not able to capture whether all infected sand flies would go on to develop high levels of infection and virulent metacyclic promastigotes. Longer incubation times are needed for parasites to undergo logarithmic replication and metacyclogenesis in the sand fly. We expect, with longer incubation times, the correlation between skin parasite load and sand fly infection level would be even stronger. Additionally, Serafim *et al.* showed sequential blood meals significantly boost parasite replication in the sand fly midgut [27]. How skin parasite burden affects parasite development in the sand fly and xenotransmission is a topic for future investigation.

It is evident high skin parasite burden significantly improves the probability of transmission to sand flies in dogs with CanL and PKDL patients, however whether skin parasite

accumulation occurs to the same extent during active human VL is not well described. Dermo-tropism of *L. infantum* during human VL has been observed, but few papers have quantified skin parasites among humans [37–39]. It has been shown that *L. infantum* VL patients co-infected with HIV develop much higher levels of parasitemia compared to immunocompetent patients, and in turn are more infectious to *Lu. longipalpis* [40]. A study of VL patients co-infected with L. *infantum*/HIV found significant cutaneous parasitism in non-lesional skin of one patient, but none of the other patients had detectable skin parasites by immunohistochem-istry [41]. Immunosuppressed individuals may develop significantly high parasite levels in peripheral blood, allowing this tissue to serve as an important viable parasite source for vector transmission.

Xenodiagnosis is the most accurate way to ascertain reservoir infectivity to vectors, however it is not trivial to perform. More easily measured correlates of xenodiagnosis need to be identi-fied for practical application. In agreement with studies in vector-infected dog cohorts, our data supports skin parasite load as a surrogate marker of dogs capable of *L. infantum* transmis-sion from vertically-infected dogs. From a public health perspective, this study reinforces and highlights the real contribution vertically infected dogs present to the infectious burden of the domestic reservoir. Within this cohort, vertically infected dogs have a basic reproductive num-ber ($R_\theta$) of ~5, which may vary by breed and litter size [4]. Vertical transmission can occur independently of vector-targeted interventions like indoor residual spraying and insecticide collars. Thus, neutering *Leishmania* infected dogs and the offspring of infected dogs regardless of diagnostic status, is critical to prevent vertical transmission and must be incorporated into *Leishmania* control strategies.

## Supporting information

**S1 Table. Cohort demographics.** Overview of the age in years, sex, and LeishVet status of xenodiagnosis cohort.
(DOCX)

**S2 Table. Cohort Bloodwork.** Overview of serum chemistry and blood count findings for dogs in each LeishVet clinical grouping. Bolded values indicate the mean is outside of the nor-mal reference range.
(DOCX)

**S1 Fig. Xenodiagnosis workflow schema.** Overview of dogs, sand flies, and sample handling workflow (left). Description of feeding and skin biopsy sites, ipsilateral pertains to side of sand fly feeding and contralateral indicates same location on opposite side where no sand fly feeding occurred.
(TIFF)

**S2 Fig. Skin parasite burden overview.** (A-B) Calculated *L. infantum* parasite burden in skin from dogs at indicated LeishVet clinical stage of disease. (A) Each dot represents one skin biopsy. 2–4 skin biopsies from each dog are shown. Kruskal-Wallis with Dunn's post-test. (B) Each dot represents one skin biopsy, separated by subject. (C) Calculated *L. infantum* parasite burden in skin from biopsies collected at either the axillary region or pinna. Each dot repre-sents the average of 1–2 skin biopsies taken at the indicated site from a single dog. Paired Wil-coxon test. Mean and standard deviation are shown for all.
(TIFF)

**S3 Fig. Calculation of *Leishmania* uptake by individual Sand Flies post xenodiagnosis.** (A) Standard curve derived from *Leishmania* parasite spiked sand flies Real Time qPCR. (B)

Parasite quantification results of all blood-meal containing female sand flies obtained after xenodiagnosis by LeishVet clinical stage. (C) No difference in mean parasite number uptake in blood fed sand flies based on anatomical feeding location via Wilcoxon paired analysis. (TIFF)

**S4 Fig. Tissue parasite burden and xenodiagnosis results in dogs with clinicopathological alterations.** Dogs within normal limits (WNL) were compared against dogs with abnormal complete blood count or serum chemistry clinicopathological findings (Abnormal). Calculated parasite burden in blood (A, n = 16), spleen (B, n = 14), average of 2–4 skin biopsies (C, n = 16). (D) Frequency of female sand flies containing a blood meal after feeding at each site. (E) Frequency of female sand flies containing $>1$ parasite equivalent 48hrs post-feeding. (F) The calculated parasite burden within sand flies 48hrs post-feeding. (A-F) Mann-Whitney test. $^{*}p<0.05$; $^{**}p<0.01$; $^{****}p<0.0001$.
(TIFF)

## Acknowledgments

We would like to thank the participating animal caretakers who donated subjects to the University of Iowa for this work. We also acknowledge the University of Iowa Comparative Pathology Core technicians for their valuable necropsy assistance. Figure artwork was created using BioRender.com.

## Author Contributions

**Conceptualization:** Breanna M. Scorza, Christine A. Petersen.

**Formal analysis:** Breanna M. Scorza.

**Funding acquisition:** Christine A. Petersen.

**Investigation:** Breanna M. Scorza, Kurayi G. Mahachi, Arin C. Cox, Angela J. Toepp, Adam Leal-Lima, Anurag Kumar Kushwaha, Patrick Kelly, Claudio Meneses, Geneva Wilson, Katherine N. Gibson-Corley, Lyric Bartholomay, Shaden Kamhawi, Christine A. Petersen.

**Methodology:** Kurayi G. Mahachi, Arin C. Cox, Angela J. Toepp.

**Project administration:** Breanna M. Scorza, Kurayi G. Mahachi, Arin C. Cox, Angela J. Toepp, Christine A. Petersen.

**Resources:** Claudio Meneses, Katherine N. Gibson-Corley, Lyric Bartholomay, Shaden Kamhawi, Christine A. Petersen.

**Supervision:** Christine A. Petersen.

**Writing – original draft:** Breanna M. Scorza.

**Writing – review & editing:** Christine A. Petersen.

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
