## [Decision Letter · Decision Letter 0]

18 May 2021

Dear Dr. Petersen,

Thank you very much for submitting your manuscript "Leishmania infantum xenodiagnosis from vertically infected dogs reveals significant skin tropism" for consideration at PLOS Neglected Tropical Diseases. As with all papers reviewed by the journal, your manuscript was reviewed by members of the editorial board and by several independent reviewers. In light of the reviews (below this email), we would like to invite the resubmission of a significantly-revised version that takes into account the reviewers' comments. 

We cannot make any decision about publication until we have seen the revised manuscript and your response to the reviewers' comments. Your revised manuscript is also likely to be sent to reviewers for further evaluation.

Sincerely,

Mitali Chatterjee

Associate Editor

Epco Hasker

Deputy Editor

Reviewer's Responses to Questions

**Key Review Criteria Required for Acceptance?**

**Methods**

-Are the objectives of the study clearly articulated with a clear testable hypothesis stated?

-Is the study design appropriate to address the stated objectives?

-Is the population clearly described and appropriate for the hypothesis being tested?

-Is the sample size sufficient to ensure adequate power to address the hypothesis being tested?

-Were correct statistical analysis used to support conclusions?

-Are there concerns about ethical or regulatory requirements being met?

Reviewer #1: The objectives of the study are articulated with a clear testable hypothesis stated.

The study design is appropriate to address the stated objectives.

Statistical analysis used to support conclusions were correct.

There is no concern about meeting ethical or regulatory requirements.

Reviewer #2: (No Response)

Reviewer #3: 1. Line 91-92 : VL is also caused by L. donovani, and author should make it clear by revising the sentence.

2. Reference 4 is about transplacental transmission in dog but not in humans. Please add appropriate reference.

3. Line 133: How did author confirm the infection?

4. Line: What is the meaning of reader value 10? Is it optical density or percent positives, and how this threshold value change with sample to sample?

5. Line 157 – 160: I guess then approx. 40-50 flies (including males) were exposed at a time during xenodiagnosis, if yes then how did this number come? This is too high number in feeding cups. Please support it with appropriate reference.

6. Line 164: Again, no reference for 30 min exposure to justify best exposure time in dogs.

7. Line 165-166: Why fed flies were incubated for 48 hrs and how this time period was established. As with many other xenodiagnosis studies, researchers use up to 72 hrs flies for outcomes. Please discuss it in detail as this is one of the important factors that can affect the outcomes.

8. Line 165-168: Is there any reason not to perform microscopy for providing direct clinal evidence?

**Results**

-Does the analysis presented match the analysis plan?

-Are the results clearly and completely presented?

-Are the figures (Tables, Images) of sufficient quality for clarity?

Reviewer #1: The analysis match the analysis plan.

The results are clearly and completely presented.

Tables and figures in the manuscript and in supporting information are of sufficient quality for clarity.

Reviewer #2: (No Response)

Reviewer #3: 9. Figure 1 do not include negative control (blood, spleen and skin) from healthy dogs and it is difficult to predict threshold without proper control. Author should make it clear in the figure legend. In addition, Figure-1 (A and B) shows the data only 14 dogs instead of 16. I would like to suggest to make schematic flow chart (as supplementary figure) showing all details e.g. recruitment, inclusion and exclusion criteria, samples collection, processing, xenodiagnosis etc. with time periods. This chart is essential to make the study process clear.

10. Figure 1A: how this 2.5 ug DNA was calculated? I guess it should be parasites genome/ml of blood. 

11. Figure 1C: Since 2 -4 skin biopsy per dog was sampled as mentioned in line: 230 therefore, please explain the figure about other biopsies data? If it is Figure S2A, then please use the colour dots representing each sample in each category.

12. Line 281- 283: This is what something strange and may be due to false positivity of DNA? Singh et al 2020 (PLoS NTD) has also discussed about this issue with PCR and thus gold standard microscopy xenodiagnosis is needed to confirm such findings. I would like to suggest to perform one set experiment in all these four categories of dog with microscopy xenodiagnosis not only in support of findings but also for direct clinical evidence with live parasite.

13. I don’t understand the need of Figure 3D. Please move it to supplementary or explain the meaning of clinical sign in context to LeishVet Guidelines stages.

14. Fig 3C-F: Please define LOD and how it was calculated?

15. Figure 3A is confusing with regard biopsy number per dogs? Is it cumulative parasite burden or one biopsy only? Please make it clear.

**Conclusions**

-Are the conclusions supported by the data presented?

-Are the limitations of analysis clearly described?

-Do the authors discuss how these data can be helpful to advance our understanding of the topic under study?

-Is public health relevance addressed?

Reviewer #1: The conclusions are supported by the data presented.

The limitations of analysis are clearly described.

The authors discuss in great detail the usefulness of the data to advance our understanding of the topic under study.

Yes, the public health relevance is addressed.

Reviewer #2: (No Response)

Reviewer #3: 16. Line 358-359: I do not agree with authors as data presented by authors are limited by qPCR. Please discuss it as one of the limitations of study and further studies with microscopy xenodiagnosis are needed to support the findings.

**Editorial and Data Presentation Modifications?**

Reviewer #1: The subject of this manuscript is within the field of interest for PNTD. It is well written and concise and therefore is suitable for publication in PNTD as Research Article, following some minor revision indicated below.

1. There is no doubt that the skin parasite load seems a better predictor than peripheral blood in canine leishmaniasis and probably in PKDL patients. However, in the case of human VL caused by L. infantum this is not so clear. Although the skin seems to play some role in immunosupressed L. donovani-infected patients it must also be taken into account that venous blood also is an important source of parasites for sand flies in the case of HIV / L. infantum coinfection. Furthermore, in the case of the human visceral leishmaniasis caused by Leishmania infantum the proportion of patients harboring parasites in their healthy skin is relatively low and such parasitism, although unusual, may be a source of infection for phlebotomine sand flies (Moura CRLP Costa CHN, Moura RD, Braga ARF, Silva VC, Costa DL. Cutaneous parasitism in patients with American visceral leishmaniasis in an endemic área. Revista da Sociedade Brasileira de Medicina Tropical. 2020; 53:e20190446. https://doi.org/10.1590/0037-8682-0446-2019). Therefore, it is very important to definitely clarify the real role of parasite burden in the blood and skin of immunocompetent and immunosuppressed individuals in the transmission of human VL.

2. Lines 330-339. I think that in the discussion addressed in this paragraph it would be very enriching also discuss the work of Mondal et al, 2018 (Mondal D, Bern C, Ghosh D, Rashid M, Molina R, Chowdhury R, Nath R, Ghosh P, Chapman LAC, Alim A, Bilbe G, Alvar J. Quantifying the infectiousness of post-kala-azar dermal leishmaniasis toward sand flies. Clinical Infectious Diseases. 2018; ciy891. https://doi.org/10.1093/cid/ciy891) jointly with the work of Singh et al, 2021 (https://doi.org/10.1016/S2666-5247(20)30166-X).

3. Why the study was not able to capture whether all infected sand flies would go on to develop high levels of infection and virulent metacyclic promastigotes?.

Reviewer #2: (No Response)

Reviewer #3: N/A

**Summary and General Comments**

Reviewer #1: This is an excellent work well-designed and conducted. The authors show in this study that canine vertical transmission of leishmaniasis should be taken into consideration in surveillance and prevention strategies. Furthermore the authors corroborate that parasite burden of dog skin can be a strong indicator of outward transmission potential.

With this paper one more piece is provided to the puzzle of the canine leishmaniasis epidemiology.

Reviewer #2: COMMENTS TO AUTHORS

There are some issues that the authors must consider. The manuscript at this moment should not be recommended or needs major revisions before to evaluate its publication.

Major comments

I suggest review the experimental design used to classify CanL disease. The authors described that CanL classification used in the study was according to Leishvet guideline. However, the stages defined in this study as "subclinical (stage 1), mild (stage 2), moderate (stage 3), or severe disease (stage 4)" are not according to the LeishVet reference. The Leishvet guideline establishes CanL disease as "stage I (mild disease), stage II (moderate disease), stage III (severe disease) and stage IV (very severe disease)". If the author adapted the CanL classification, the used criteria should be described. Since small sample was evaluated in this study, I suggest the authors consider the reclassification of dogs according to "no clinicopathological alterations versus with clinicopathological alterations".

The authors describe that skin parasite burden is more important than dog´s clinical status. I suggest to reword the sentences with these statements, since LeishVet stage 3 and 4 had few dogs, 2 and 1, respectively. Their emphasized does not appear appropriate.

In Material and methods section should provide more information about dogs sampling. What is their origin? How was performed the follow up of these dogs since birth? It is not clear if after the birth, the dogs of this study have never been exposed to areas with potential risk for L. infantum infection transmitted by the vector. 

In figure 1, is not clear how statistical analysis was performed. How was performed each group comparison? For example, parasite burden in blood was significantly higher compared with what? Line 215-216 in the statement L. infantum parasite burden in (A) blood (p=0.004; n=16), (B) spleen (p=0.153; 216 n=14), or (C) skin (p=0.072; n=16), these P value refers to comparison between blood and spleen or skin..., please clarify. I suggest include in figure 1 another graph to show parasite burden per tissue without CanL stages.

Minor comments

I suggest using the term "canine leishmaniasis (CanL)” when authors refer the dog disease.

Line 97. References (21, 22) are not correctly numbered in the order that they appear in the text.

Line 127 change ‘confirms’ to ‘suggests’

Line 287 change ‘subclinical disease’ to subclinical infection

In table S1, I suggest to provide age class frequency (young, adult or elderly). Exclude ‘mean (SD)’. Also, I suggest include information on breed.

In table S2, I suggest to include the reference source which was take out the value normal range of the laboratory data.

Lines 172-173 the statement “Skin biopsies were taken from axillary and pinnal sand fly feeding sites as well as contralateral”, I suggest to provide more information on number fragment tissues were obtained from each skin region.

Line 292-293. On the statement “the age range of dogs with the highest average parasite uptake per sand fly in this cohort was 3-4 years old”, what is hypothesis for this finding?

Line 305-306. In the statement “Parasitemia is known to correlate with Leishmania parasite transmission to sand flies”, provide reference.

Line 360-362. The statement “Moderately diseased dogs (LeishVet levels 2-3) were the most infectious to Lu. longipalpis sand flies compared to dogs with mild (LeishVet level 1) or severe disease (LeishVet level 4)” is confused. It is not accordign to CanL classification used by authors (line 151-152).

Line 394-396 statement “Our data supports skin parasite load as a surrogate marker of dogs with L. infantum transmission potential from both horizontally and vertically infected dogs”, I suggest exxclude “horizontally”, since this study not compared parasite skin load from horizontally infected dogs.

Reviewer #3: The study by Scorza et al have conducted xenodiagnosis on cohort of dogs to examined infectiousness of vertically infected dogs with L.infantum at varying stages of clinical diseases.

PLOS authors have the option to publish the peer review history of their article (what does this mean?). If published, this will include your full peer review and any attached files.

Reviewer #1: No

Reviewer #2: No

Reviewer #3: No
---

## [Decision Letter · Decision Letter 1]

24 Sep 2021

Dear Dr. Petersen,

We are pleased to inform you that your manuscript 'Leishmania infantum xenodiagnosis from vertically infected dogs reveals significant skin tropism' has been provisionally accepted for publication in PLOS Neglected Tropical Diseases.

Best regards,

Mitali Chatterjee

Associate Editor

Epco Hasker

Deputy Editor

Reviewer's Responses to Questions

**Key Review Criteria Required for Acceptance?**

**Methods**

-Are the objectives of the study clearly articulated with a clear testable hypothesis stated?

-Is the study design appropriate to address the stated objectives?

-Is the population clearly described and appropriate for the hypothesis being tested?

-Is the sample size sufficient to ensure adequate power to address the hypothesis being tested?

-Were correct statistical analysis used to support conclusions?

-Are there concerns about ethical or regulatory requirements being met?

Reviewer #3: Yes

**Results**

-Does the analysis presented match the analysis plan?

-Are the results clearly and completely presented?

-Are the figures (Tables, Images) of sufficient quality for clarity?

Reviewer #3: Yes

**Conclusions**

-Are the conclusions supported by the data presented?

-Are the limitations of analysis clearly described?

-Do the authors discuss how these data can be helpful to advance our understanding of the topic under study?

-Is public health relevance addressed?

Reviewer #3: Yes

**Editorial and Data Presentation Modifications?**

Reviewer #3: Authors have substantially modified the draft and now it is more clear.

**Summary and General Comments**

Reviewer #3: I would like to suggest to add Flow diagram of experimental design to make the process including exclusion and inclusion criteria clear!

PLOS authors have the option to publish the peer review history of their article (what does this mean?). If published, this will include your full peer review and any attached files.

Reviewer #3: No

---

## [Editor Report · Acceptance letter]

4 Oct 2021

Dear Dr. Petersen,

We are delighted to inform you that your manuscript, "Leishmania infantum xenodiagnosis from vertically infected dogs reveals significant skin tropism," has been formally accepted for publication in PLOS Neglected Tropical Diseases.

Best regards,

Shaden Kamhawi

co-Editor-in-Chief

Paul Brindley

co-Editor-in-Chief
